# Circulating Tumor DNA as a Novel Biomarker Optimizing Chemotherapy for Colorectal Cancer

**DOI:** 10.3390/cancers12061566

**Published:** 2020-06-13

**Authors:** Hiroki Osumi, Eiji Shinozaki, Kensei Yamaguchi

**Affiliations:** Department of Gastroenterology, Cancer Institute Hospital, Japanese Foundation for Cancer Research, Tokyo 135-8550, Japan; hiroki.osumi@jfcr.or.jp (H.O.); kensei.yamaguchi@jfcr.or.jp (K.Y.)

**Keywords:** circulating tumor DNA, liquid biopsy, colorectal cancer, minimal residual disease, and biomarker

## Abstract

Liquid biopsy is a minimally invasive method for detecting soluble factors, including circulating tumor DNA (ctDNA), in body fluids. ctDNA carrying tumor-specific genetic or epigenetic alterations is released into circulation from tumor cells. ctDNA in the plasma contains somatic mutations that have occurred in the tumor, and reflects tumor progression and therapeutic effects promptly and accurately. Furthermore, ctDNA is useful for early detection of recurrence and estimation of prognosis and may be utilized for diagnosis and personalized medicine for treatment selection. Thus, in the near future, it will be possible to select the most appropriate treatment based on real-time genetic information using ctDNA.

## 1. Liquid Biopsy and Circulating Tumor DNA

It is important to identify tissue-based biomarkers before treatment to predict the prognosis and efficacy of systemic therapy in metastatic colorectal cancer (mCRC) [1]. For, example, *KRAS* and *NRAS* mutations [2,3,4] and HER2 amplification [5,6] are negative predictive factors of efficacy for epidermal growth factor receptor (EGFR) inhibitor in mCRC. *BRAF V600E* mutation is a strong predictive factor of poor prognosis and negative predictor of standard chemotherapy [7,8]. In contrast, HER2-targeted therapies are an effective approach for HER2-positive mCRC [9,10]. Furthermore, microsatellite instability is a predictive factor of efficacy for immune checkpoint inhibitors in mCRC [11,12]. Although tissue biomarkers are most useful in estimating prognosis and predicting therapeutic response, a new diagnostic concept referred to as ‘liquid biopsy’ has drawn attention over the past few years [13,14]. Liquid biopsy is defined as a minimally invasive method for detecting several soluble factors such as circulating tumor DNA (ctDNA), circulating tumor cells, and exosomes using body fluids [15,16,17]. Among them, the studies of ctDNA are rapidly increasing owing to the advancement of molecular technology that facilitates the detection and quantification of tumor-associated genomic variants. Cell-free DNA (cfDNA) is extracellular DNA present in a number of body fluids including blood, urine [18], cerebrospinal fluid [19], pleural fluid [20], ascites [21], and saliva [22]. It is derived from both normal and cancer cells undergoing apoptosis and necrosis [23]. cfDNA concentrations may change according to comorbidities such as cerebrovascular disorder [24], or physical condition [25]. cfDNA clearance rapidly occurs in several organs such as the kidney, liver and spleen and their half life time is very short, approximately from several minutes to hours [26,27,28]; these data suggest that cfDNA analysis may represent a real-time tumor burden. ctDNA defines the fraction of cfDNA that originates from tumor cells, generally inferred by the detection of somatic variants, and are a small fraction of the total cfDNA that contains point mutations, rearrangements, amplifications, and even gene copy number variations. The comparison between conventional biopsy and liquid biopsy reported by the joint review of ctDNA by the American Society of Clinical Oncology/College of American Pathologists [29] is shown in Appendix A. A liquid biopsy can deliver more complete information regarding the patient’s entire tumor burden, because the sample represents all tumor DNA present in the circulation as opposed to the spatial limitations of a biopsy sample from a single lesion within a single anatomic site [30]. To understand the clinical significance of ctDNA, investigation of the relationship between clinicopathological factors and ctDNA is of importance. The sensitivity and specificity of ctDNA detection in plasma has been correlated with tumor size and stage in CRC [31,32]. Furthermore, the significant association between ctDNA and metastatic organs, particularly in liver metastasis, has been suggested [31,33]; however, no relationship has been observed in peritoneal metastasis [31,33] and lung metastasis in CRC [31,34]. Thus, it is necessary to take into account the stage and patient background factors in order to interpret ctDNA results accurately.

## 2. Advancement of Detection Systems of ctDNA

ctDNA includes DNA alterations that are tumor-specific genetic abnormalities (for example: point mutations, deletions/insertions), epigenetic alterations (for example: methylation of tumor suppressor gene promoter), and loss of heterozygosity [23,35]. The high concordance of *RAS/BRAF* mutational status between ctDNA and tumor tissue based on examination has been reported in mCRC [33,34,36]. The methodologies used for the detection of ctDNA are shown in Appendix A [37,38,39,40,41,42,43,44,45,46,47]. Generally, real-time PCR methods that apply mutation-specific PCR have been used to detect plasma *RAS/BRAF* mutation. Because new technologies such as BEAMing (beads, emulsion, amplification, and magnetics) [41] and droplet digital-PCR (dPCR) [32] have been developed, the detection sensitivity of ctDNA has improved greatly. However, PCR-based detection methods enable the detection of only a few known mutations [48]. The detection sensitivity of the Sanger method, a first generation sequencing method, is very low at about 10% and hence not suitable for ctDNA detection. The detection sensitivity of next-generation sequencing (NGS) with deep sequencing has increased to 0.1‒1.0% [42,44]. In ctDNA analysis using NGS, it is possible to detect newly emerged mutations after treatment, and evaluate serial changes in the tumor genome and the mechanism of treatment resistance. On the other hand, NGS has been reported to have higher PCR error and misreading than dPCR, and the high cost is also a challenge in clinical application [49]. Therefore, dPCR and NGS should be used according to the clinical or research purpose and after understanding their characteristics. Furthermore, as one of the latest developments in the methodology of ctDNA, the PCR assay panel is designed based on phylogenetic analysis, targeting clonal and subclonal single nucleotide variants to facilitate non-invasive tracking of the patient-specific tumor phylogeny [50]. Using this technology, a multicenter cohort study has shown that, in multivariate analysis, ctDNA status is independently associated with relapse after adjusting for known clinicopathological risk factors in stage I to III CRC [51]. Therefore, because this technology uses tissue DNA obtained from individual patients, it may contribute further to personalized medicine compared to conventional gene panels in CRC.

## 3. Clinical Utility of ctDNA for Advanced CRC

ctDNA analysis can be useful to detect cancer at an early stage in patients, to detect minimal residual disease (MRD), and to detect early relapse after curative surgery [52]. Furthermore, it can also be useful to predict early chemotherapeutic responses and to monitor secondary resistance of chemotherapy in patients with metastasis [52]. In this section, we review the clinical utility of ctDNA for MRD/recurrence monitoring, prediction of chemotherapeutic response, and prediction of secondary resistance to chemotherapy in patients with CRC.

### 3.1. Minimal Residual Disease and Recurrence Monitoring

The purpose of postoperative adjuvant chemotherapy is to reduce the risk of recurrence and improve prognosis. To date, the indication for postoperative adjuvant chemotherapy is based on pathological biomarkers, such as invasion depth (T factor) and the presence or absence of lymph node metastasis (N factor). Postoperative fluoropyrimidine and oxaliplatin combination therapy is recommended as the standard treatment for stage III patients and stage II patients with a high risk of recurrence, but not for other stage II patients [53,54], although there is not enough evidence to support this classification. Furthermore, although previous reports showed that several biomarkers predict the efficacy of adjuvant chemotherapy, such as microsatellite instability status [55,56,57], 18q loss of heterozygosity [58,59], and distinct gene signatures [60,61,62], among others, the development of molecular markers that more accurately predict the risk of recurrence is still extremely urgent, as they have not yet been used in clinical practice.

#### 3.1.1. Previously Reported Clinical Relevance of ctDNA for Adjuvant Chemotherapy

ctDNA analysis can be useful to detect in patients cancer at an early stage, to detect MRD, and to detect early relapse after curative surgery. Detection of ctDNA after curative surgery may signal the presence of MRD even in the absence of any other clinical evidence of disease (Figure 1) [63,64].

Liquid biopsy approaches might be suitable for estimating MRD, because residual tumor components can be detected using high sensitivity methods. Tie et al. [63] reported a prospective cohort study in patients with stage II colon cancer after curative resection to evaluate clinical utility of postoperative ctDNA status about MRD detection. ctDNA was analyzed using the Safe-SeqS method, which is a low error rate sequencing technology. A unique identifier is assigned to each DNA molecule to be analyzed using amplification and deep sequencing of universally tagged DNA molecules. This allows for differentiation between real mutations and errors introduced during the amplification and sequencing processes. Postoperative ctDNA was detected in 14 of 178 cases (7.9%); of the 14 cases, 11 cases relapsed (78.6%). In contrast, recurrence was observed in 16 of 164 cases (9.8%), where ctDNA was negative after surgery. The postoperative ctDNA-positive cases had significantly shorter recurrence free survival (RFS) (HR 18.0, *p* < 0.001) than the negative cases, and the 3-year RFS rate were 0% for ctDNA-positive and 90% for negative cases. Further, postoperative ctDNA status was an independent prognostic factor for RFS in multivariate analysis, with or without adjuvant therapy [65]. Tie et al. [66] also reported on the clinical utility of ctDNA for MRD in patients with locally advanced rectal cancer (T3/T4 and/or lymph node metastasis-positive). ctDNA was analyzed using Safe-SeqS methods. In their study, 23 cases (15.0%) relapsed and although there was no association between pre-treatment ctDNA levels and RFS, patients who were positive for ctDNA after chemoradiotherapy also had significantly shorter RFS (post-chemoradiotherapy, HR 6.6, *p* < 0.001, after surgery: HR 13.0, *p* < 0.001). Three-year RFS rate was 33.0% for ctDNA-positive cases and 87.0% for negative cases after surgery. Postoperative ctDNA status was an independent prognostic factor for RFS in multivariate analysis irrespective of postoperative adjuvant therapy (with adjuvant therapy: HR 10.0, *p* < 0.001; without adjuvant therapy: HR 22.0, *p* < 0.001). Another study also reported the results of the clinical utility of ctDNA as a biomarker for the detection of MRD, identification of patients at a high risk of recurrence, and early detection of recurrence by longitudinal ctDNA analysis in patients with stage I to III CRC [51].

These results suggest that ctDNA may be useful for assessing the risk of postoperative recurrence, monitoring the efficacy of adjuvant chemotherapy, and as a biomarker for the early detection of recurrence in patients with CRC after curative resection. ctDNA may improve the treatment outcomes by helping administer a more intensive treatment to patients with MRD at an increased risk of relapse. Furthermore, ctDNA may also contribute to preventing unnecessary comorbidities and adverse events, because physicians can easily select patients who are likely to benefit from adjuvant chemotherapy. Currently, clinical trials stratifying treatments according to ctDNA status are ongoing, and further novel findings regarding ctDNA for MRD are expected.

#### 3.1.2. Prospective Trials to Validate the Clinical Utility of ctDNA for Adjuvant Chemotherapy

Planned clinical trials to validate clinical utility of ctDNA for adjuvant chemotherapy are summarized in Appendix A. Among them, the DYNAMIC-III trial (ACTRN 12617001566325) is a multicenter phase II/III randomized controlled study to compare the standard treatment with treatment based on the result of ctDNA analysis in stage III colon cancer after curative resection. The purpose of the DYNAMIC-III study is to determine whether chemotherapy based on the presence or absence of ctDNA after curative surgery for stage III CRC is more effective than standard of care treatment. In this trial, subjects will be randomly assigned 1:1 between the standard and experimental treatment groups based on ctDNA information obtained 5–6 weeks after surgery with curative resection for stage III colon cancer. In the group that determines treatment selection based on the results of ctDNA, if ctDNA is positive, treatment intensity is escalated, while if ctDNA is negative, treatment intensity is de-escalated or the treatment period is shortened. The endpoints of this study are as follows: (i) a de-escalation treatment strategy is non-inferiority in terms of the rate of 3-year RFS and (ii) an escalation treatment strategy is superior in the 24 months RFS compared to the standard of care treatment. Second, IMPROVE Intervention Trial Implementing Non-Invasive Circulating Tumor DNA Analysis to Optimize the Operative and Postoperative Treatment for Patients with CRC (ClinicalTrials.gov Identifier: NCT03748680) is also currently ongoing. The purpose of the IMPROVE Intervention Trial is to evaluate the efficacy and safety of adjuvant chemotherapy for ctDNA positive patients after curative surgery in stage I/II CRC. The endpoint of this study is to evaluate whether adjuvant chemotherapy (FOLFOX or CapeOx) improves the disease free survival (DFS) in patients with MRD (ctDNA positive) to whom adjuvant chemotherapy has not been recommended as the standard treatment. Finally, the CIRCULATE study is an investigator-initiated, multicenter; prospective, randomized, controlled trial (ClinicalTrials.gov Identifier: NCT04089631), which is also ongoing. The purpose of the CIRCULATE study is to validate the efficacy of adjuvant therapy in patients with stage II colon cancer. The endpoint of this study is to compare the DFS of patients who are positive for postoperative ctDNA, with and without the administration of capecitabine. Based on the above data, blood-based ctDNA testing is expected to update the current primary tumor/regional lymph nodes/metastasis (TNM) classification of cancer staging. The development of new TNM staging, the so-called “TNM and Blood: TNMB system” may outperform the current TNM cancer staging system.

### 3.2. Treatment Response, Clonal Evolution, and Resistance

Liquid biopsies may play an important role in the monitoring the treatment response and/or resistance to systemic therapy (Figure 2) [65,67].

The clinical relevance of ctDNA analysis for monitoring the therapeutic response has been reported in patients with mCRC [67,68,69]. High basal ctDNA levels were associated with a short overall survival, and ctDNA assessment could act an early surrogate marker of treatment response [70,71]. Early changes in ctDNA levels during chemotherapy with molecular targeted drugs can predict the treatment outcomes [65,72,73,74]. Similarly, liquid biopsies have been used to identify mechanisms of resistance to EGFR inhibitor therapy in patients with mCRC [75]. Interestingly, the emergence of resistant *KRAS* mutated clones could be detected up to several months before radiological evidence of disease progression [76,77]. Mutations in the EGFR extracellular domain that negate the binding of EGFR inhibitors, such as cetuximab and panitumumab, have been detected in patients with CRC who had achieved a partial response or stable disease after EGFR inhibitor therapies [78]. Mesenchymal-epithelial transition amplification, which is reported as one of the resistance factors of EGFR blockade, has also been detected in the plasma of patients with mCRC after initiation of EGFR inhibitor therapy [79]. Furthermore, Russo et al. [80] longitudinally monitored plasma ctDNA to assess *LMNA*–*NTRK1* gene fusion status in a patient with mCRC during treatment with the tyrosine kinase receptor inhibitor, entrectinib. Information obtained from liquid biopsies has enabled us to detect previously unknown mutations related to resistance to entrectinib [80]. These findings suggest the potential of ctDNA analyses for monitoring of clonal evolution and guiding therapeutic decisions. Serial ctDNA analysis in patients with CRC demonstrated elevation of mutant *KRAS* clones during anti-EGFR therapy and a later decline in the mutant *KRAS* clones upon the withdrawal of anti-EGFR therapy [67]. Some studies reported that the emergence of resistant *KRAS* mutated clones during anti-EGFR therapy influenced the clinical outcomes of patients treated with anti-EGFR therapy. Cremolini et al. [81] reported that patients without plasma *RAS* mutations showed a partial response; in contrast, patients with plasma *RAS* mutations did not exhibit a partial response. Furthermore, patients without plasma *RAS* mutation had significantly longer progression free survival (PFS) than those with plasma *RAS* mutation, with a median PFS of 4.0 vs. 1.9 months (*p* = 0.03) in the CRIKET cetuximab re-challenge trial [81]. These results suggest that screening of EGFR signaling pathway by liquid biopsy may contribute to select patients that would benefit from anti-EGFR antibody re-challenge. As evidence of successful re-challenge strategies with targeted therapies may be found in patients with other tumor types, such as melanoma or non-small cell lung cancer, it is necessary to verify the re-challenge strategy of anti-EGFR antibody using ctDNA for mCRC.

## 4. Conclusions

ctDNA is a promising biomarker that can obtain quantitative and qualitative comprehensive tumor DNA in a minimally invasive manner. ctDNA assays will guide the choice of the most appropriate chemotherapy treatments, and predict early recurrence, chemotherapeutic response, and resistance. However, currently, most of them are only being used in the context of clinical trials. Thus, as ctDNA analysis is expected to be a useful decision-making tool that would contribute to precision medicine, further data collection from prospective studies is required to definitely establish ctDNA analysis as a future clinical application.

## Figures and Tables

**Figure 1 cancers-12-01566-f001:**
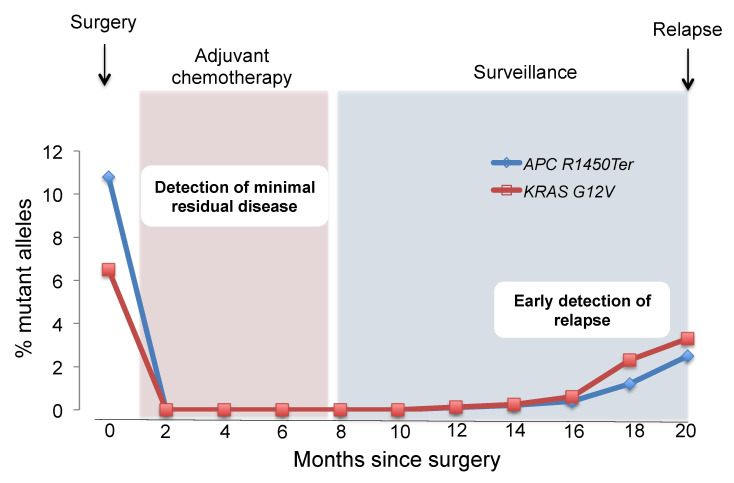
Minimal residual disease (MRD) monitoring and early diagnosis of relapse. Liquid biopsy approaches might be well suited to measuring MRD, as residual tumor components can be detected with high sensitivity. Because MRD-positive patients have higher risk of relapse, early diagnosis of relapse may improve patient outcomes. This figure reflects a representative case of our own original data.

**Figure 2 cancers-12-01566-f002:**
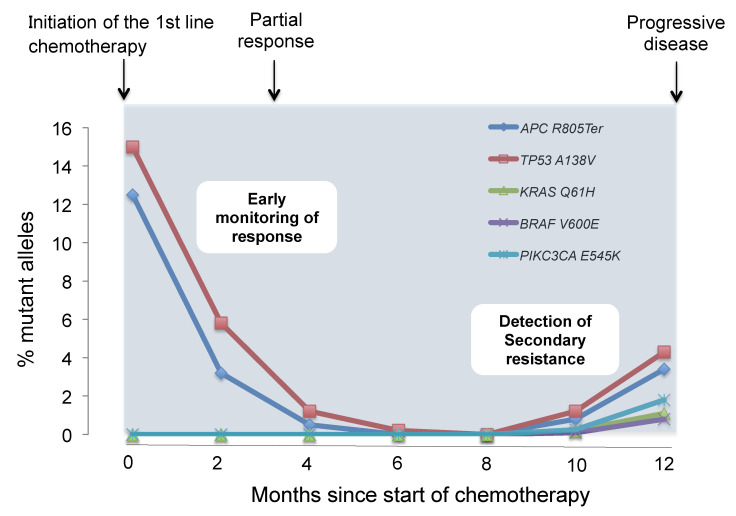
Evaluation of chemotherapeutic response and prediction of resistance to chemotherapy. Serial tumor-specific mutation analysis in the blood of patients is useful to monitor response and resistance to molecular targeted drugs. Early decrease of mutant allele after chemotherapy may predict better therapeutic response and clinical outcomes. Furthermore, liquid biopsies can detect the emergence of resistant clones of chemotherapy before radiographic confirmation of disease progression. This figure reflects a representative case of our own original data.

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
