# Peer review of "Circulating Tumor DNA as a Novel Biomarker Optimizing Chemotherapy for Colorectal Cancer"

_cancers, 2020, doi:10.3390/cancers12061566_

Round 1

Reviewer 1 Report

The authors provide a review of current literature around the use of circulating tumor DNA as a biomarker for optimizing chemotherapy in colorectal cancer. They have included a broad range of relevant literature with reference to recent clinical trials. 

After reviewing the manuscript, I have provided some suggestions below.

Section 3.1.1 seems out of place in this section and should be moved, or removed from the manuscript entirely. This section does not provide a comprehensive overview of other candidate biomarkers for adjuvant chemotherapy (e.g. DPD, UGT1A1*28, CEA, 18qLOH etc) and this should be addressed either by acknowledging this or expanding this section.  

Figures 1 and 2 describe the use of MRD and relapse, and prediction of resistance to chemotherapy, respectively. Can the authors provide some insight as to what data was used to generate these figures? Is this based on modelling? Or actual patient data?

Figure 2 should read "Months since start of chemotherapy". Query as to the positioning of "Partial response" - is this meant to align with a specific time point i.e. 2 months?

Below are some suggested corrections with the corresponding line item:

41 "...and these data suggest.." To which data are the authors referring?

47 "In theory..." I would suggest that this is no longer considered a theory, but there is supporting evidence to substantiate this statement.

82 The shift in discussion of ctDNA to cfDNA and back again is difficult to follow through this section. Please amend.

94 "...stage II patients with a high recurrence rate..." I believe this should read stage II patients with a high risk of recurrence.

108 5FU/LV + Oxaliplatin is defined as FLOX, but line 111 refers to FOLFOX, which has not been predefined but is presumably the same regimen?

131 Should read "SUNRISE"

135 "These data suggest that assays based on multiple gene expressions are confirmed to be useful in predicting recurrence risk." I think it suggests this could be useful in predicting risk, but this has not been adopted in clinical practice to my knowledge.

149 No need to reference [57] again here as it was referenced in the previous sentence.

205 Define "TNMB".

223 Consider changing to "radiological evidence of disease progression."

226 "MET" not previously defined.

236 Should read "Clemolini et al..."

238 Remove capital "T" following "Furthermore,"

239 Check nomenclature in this section - sometimes RAS is italicized but sometimes it is not?

242 I am not sure if the phrase "..enrich patients.." is suitable here. Consider revising.

I believe the manuscript would benefit from including the following additional references-

Parikh et al 2020 doi: 10.1158/1078-0432.CCR-19-3467

Messaoudi et al 2016 doi 10.1158/1078-0432.CCR-15-0297

Author Response

Reviewer: 1 
Comments to the Author 

The authors provide a review of current literature around the use of circulating tumor DNA as a biomarker for optimizing chemotherapy in colorectal cancer. They have included a broad range of relevant literature with reference to recent clinical trials. After reviewing the manuscript, I have provided some suggestions below.

Comment 1:

Section 3.1.1 seems out of place in this section and should be moved, or removed from the manuscript entirely. This section does not provide a comprehensive overview of other candidate biomarkers for adjuvant chemotherapy (e.g. DPD, UGT1A1*28, CEA, 18qLOH etc) and this should be addressed either by acknowledging this or expanding this section.  

Response to Comment 1:

As suggested by the reviewer, we combined the sentences 3.1 and 3.1.1 and shortened them (Page 3, Line 91-102).

Comment 2:

Figures 1 and 2 describe the use of MRD and relapse, and prediction of resistance to chemotherapy, respectively. Can the authors provide some insight as to what data was used to generate these figures? Is this based on modelling? Or actual patient data?

Response to Comment 2: 

Figures1 and 2 consist of our original data(Page 4, Line 111-112, Page 6, Line 185-186).

Comment 3:

Figure 2 should read "Months since start of chemotherapy". Query as to the positioning of "Partial response" - is this meant to align with a specific time point i.e. 2 months?

Response to Comment 3:

As suggested by the reviewer, we revised the title of X-axis of Figure1 and 2. Positioning of "Partial response" is located in first radiologic evaluation (around 2-3 months after start of chemotherapy).

Comment 4:

Below are some suggested corrections with the corresponding line item:

41 "...and these data suggest.." To which data are the authors referring?

Response to Comment:

We referred three original papers and added them in the manuscript(Page 1, Line 41).

47 "In theory..." I would suggest that this is no longer considered a theory, but there is supporting evidence to substantiate this statement.

Response to Comment:

I agreed with your onion. I deleted this sentence.

82 The shift in discussion of ctDNA to cfDNA and back again is difficult to follow through this section. Please amend.

Response to Comment:

I agreed with your onion. I deleted this sentence about cfDNA.

94 "...stage II patients with a high recurrence rate..." I believe this should read stage II patients with a high risk of recurrence.

Response to Comment:

As we combined the sentences 3.1 and 3.1.1 and shortened them, we delete this sentence in the manuscript.

108 5FU/LV + Oxaliplatin is defined as FLOX, but line 111 refers to FOLFOX, which has not been predefined but is presumably the same regimen?

Response to Comment:

As we combined the sentences 3.1 and 3.1.1 and shortened them, we delete this sentence in the manuscript.

131 Should read "SUNRISE"

Response to Comment:

As we combined the sentences 3.1 and 3.1.1 and shortened them, we delete this sentence in the manuscript.

135 "These data suggest that assays based on multiple gene expressions are confirmed to be useful in predicting recurrence risk." I think it suggests this could be useful in predicting risk, but this has not been adopted in clinical practice to my knowledge.

Response to Comment:

As we combined the sentences 3.1 and 3.1.1 and shortened them, we delete this sentence in the manuscript.

149 No need to reference [57] again here as it was referenced in the previous sentence.

Response to Comment:

I agreed with your onion. I deleted this sentence.

205 Define "TNMB".

Response to Comment:

According to the comments by reviewer, we revised the sentence as follows; “TNM and Blood: TNMB system,”(Page 5, Line 173).

223 Consider changing to "radiological evidence of disease progression."

Response to Comment:

According to the comments by reviewer, we revised this sentence

(Page 6, Line 194-195).

226 "MET" not previously defined.

Response to Comment:

According to the comments by reviewer, According to the comments by reviewer, we revised this sentence as follows; MET: Mesenchymal-epithelial transition(Page 6, Line 198).

236 Should read "Clemolini et al..."

Response to Comment:

According to the comments by reviewer, we revised this sentence

(Page 6, Line 209).

238 Remove capital "T" following "Furthermore,"

Response to Comment:

I agreed with your onion. I deleted this sentence (Page 6, Line 211).

239 Check nomenclature in this section - sometimes RAS is italicized but sometimes it is not?

Response to Comment:

According to the comments by reviewer, we revised these sentences.

242 I am not sure if the phrase "..enrich patients.." is suitable here. Consider revising.

Response to Comment:

According to the comments by reviewer, we revised this sentence as follows; These results suggest that screening of EGFR signaling pathway by liquid biopsy may contribute to select patients that would benefit from anti-EGFR antibody re-challenge (Page 6, Line 213-215).

I believe the manuscript would benefit from including the following additional references-

Response to Comment:

According to the comments by reviewer, we added references in the manuscript (Page 6, Line 190, Line 192).

Reviewer 2 Report

In this review of the liquid biopsy concept and its potential for clinical use, the authors focus on the assessment of the two crucial points as regards the systemic treatment of colorectal cancer patients. The first point is the indication for chemotherapy, for which the authors provide an updated review of the scientific background and describe very relevant ongoing clinical trials. In the second section, they report on the potential use of the liquid biopsy as a tool for efficacy monitoring of systemic chemotherapy. The second section (3.2.) is however significantly shorter than the first one (3.1.) which makes the whole article somewhat unbalanced. Very short conclusion section ads to that impression in addition I find the transition from the main body of the article to the conclusion section not very fluent.

Author Response

Reviewer: 2 
Comments to the Author 

In this review of the liquid biopsy concept and its potential for clinical use, the authors focus on the assessment of the two crucial points as regards the systemic treatment of colorectal cancer patients. The first point is the indication for chemotherapy, for which the authors provide an updated review of the scientific background and describe very relevant ongoing clinical trials. In the second section, they report on the potential use of the liquid biopsy as a tool for efficacy monitoring of systemic chemotherapy. The second section (3.2.) is however significantly shorter than the first one (3.1.) which makes the whole article somewhat unbalanced. Very short conclusion section ads to that impression in addition I find the transition from the main body of the article to the conclusion section not very fluent.

Response to Comment:

According to the comments by reviewer,we combined the sentences 3.1 and 3.1.1 and shortened them (Page 3, Line 91-102).

Furthermore, we modified the description in the conclusion based on the contents of the manuscript as follows; ctDNA is a promising biomarker that can obtain quantitative and qualitative comprehensive tumor DNA in a minimally invasive manner. ctDNA assays will guide the choice of the most appropriate chemotherapy treatments, and predict early recurrence, chemotherapeutic response and resistance. However, currently, most of them are only being used in the context of clinical trials. Thus, as ctDNA analysis is expected to be a useful decision-making tool that would contribute to precision medicine, further data collection from prospective studies is required to definitely establish ctDNA analysis as a future clinical application (Page 7, Line 220-226).

Reviewer 3 Report

The authors review putative roles for ctDNA assay in management of CRC. This non-invasive method yields information re. tumour recurrence and response to adjuvant and palliative chemotherapy. This is a useful topic for review.                                             

Section 1: The authors should review issues limiting ctDNA assays. It would be useful to provide an overview of sensitivity and specificity. What is the sampling error, what tumour size is the estimated lower limit for ctDNA detection? Since ctDNA derives from dying cancer cells it may not represent dominant subclones which are resistant to chemotherapy. 

Table 1 should be Table S1.    The citation for this information should be earlier in the sentence as follows “…American Pathologists [26] is shown in Table S1.

Section 2: The authors might include a table presenting available methods including BEAMing and droplet digital-PCR, with data for detection sensitivity (see above).                 

Section 3:

3.1 Risk factors for mets following adjuvant chemotherapy are well known and should not be listed in full.

Similarly, Section 3.1.1 could be shortened.

The later parts of Section 3 are informative with good reference to the literature.  

Section 3.1.3, Table 2 should be Table S2. 

4: The conclusion does not follow on from Section 3.

“Along with the advancement of detection technology,  improvement in sensitivity and cost reduction would be necessary for clinical application”

 The earlier section suggests ctDNA assays will guide management decisions and predict early recurrence, and currently it is being used within clinical trials. 

Author Response

Reviewer: 3
Comments to the Author 

The authors review putative roles for ctDNA assay in management of CRC. This non-invasive method yields information re. tumour recurrence and response to adjuvant and palliative chemotherapy. This is a useful topic for review.             

Comment 1:                           

Section 1: The authors should review issues limiting ctDNA assays. It would be useful to provide an overview of sensitivity and specificity. What is the sampling error, what tumour size is the estimated lower limit for ctDNA detection? Since ctDNA derives from dying cancer cells it may not represent dominant subclones which are resistant to chemotherapy. 

Response to Comment 1:

According to the comments by reviewer, we added several sentences about detection sensitivity and specificity of ctDNA in plasma in the Section1 (Page 2, Line 50-56).

Comment 2:           

Table 1 should be Table S1. The citation for this information should be earlier in the sentence as follows “…American Pathologists [26] is shown in Table S1.

Response to Comment 2:

According to the comments by reviewer, we revised the description of table and citation (Page 2, Line 46-47).

Comment 3: 

Section 2: The authors might include a table presenting available methods including BEAMing and droplet digital-PCR, with data for detection sensitivity (see above).                 

Response to Comment 3:

According to the comments by reviewer, we added methodologies for detecting ctDNA with detection sensitivity in Table S2 (Page 2, Line 62-63, Page7, Line 228-229, Table S2).

Comment 3:       

Section 3: 3.1 Risk factors for mets following adjuvant chemotherapy are well known and should not be listed in full. Similarly, Section 3.1.1 could be shortened. 

Response to Comment 3:

As suggested by the reviewer, we combined the sentences 3.1 and 3.1.1 and shortened them (Page 3, Line 91-102).

Comment 4:   

The later parts of Section 3 are informative with good reference to the literature.  Section 3.1.3, Table 2 should be Table S2. 

Response to Comment 4:

Thank you for your suggestion. According to the comments by reviewer, we revised the description of the table.

Comment 5:   

4: The conclusion does not follow on from Section 3.

“Along with the advancement of detection technology, improvement in sensitivity and cost reduction would be necessary for clinical application”

The earlier section suggests ctDNA assays will guide management decisions and predict early recurrence, and currently it is being used within clinical trials. 

Response to Comment 5:

According to the comments by reviewer, we revised the description in the conclusion as follows; ctDNA is a promising biomarker that can obtain quantitative and qualitative comprehensive tumor DNA in a minimally invasive manner. ctDNA assays will guide the choice of the most appropriate chemotherapy treatments, and predict early recurrence, chemotherapeutic response and resistance. However, currently, most of them are only being used in the context of clinical trials. Thus, as ctDNA analysis is expected to be a useful decision-making tool that would contribute to precision medicine, further data collection from prospective studies is required to definitely establish ctDNA analysis as a future clinical application (Page 7, Line 220-226).

Round 2

Reviewer 1 Report

The authors have undertaken significant revision of the manuscript and have improved readability and flow/balance of content. They have taken into account the suggestions from each of the reviewers and have presented a revised manuscript that is both comprehensive and well organized.